# Favorable Nonclinical Safety Profile of RSVpreF Bivalent Vaccine in Rats and Rabbits

**DOI:** 10.3390/vaccines13010026

**Published:** 2024-12-31

**Authors:** Jun Zhou, Christopher J. Bowman, Vicki R. Markiewicz, Balasubramanian Manickam, Emily Gomme, Rani S. Sellers, Cynthia M. Rohde

**Affiliations:** 1Drug Safety Research and Development, Pfizer Research & Development, Groton, CT 06340, USA; christopher.j.bowman@pfizer.com (C.J.B.);; 2Independent Researcher, East Lyme, CT 06333, USA; 3Clinical Immunology and High-Throughput Assays, Vaccine Research and Development, Pfizer Research & Development, Pearl River, NY 10965, USA; 4Drug Safety Research and Development, Pfizer Research & Development, Pearl River, NY 10965, USA

**Keywords:** respiratory syncytial virus, vaccine, toxicity, pregnancy, fertility, developmental toxicity, maternal immunization

## Abstract

**Background**: Respiratory syncytial virus (RSV) infections usually cause mild, cold-like symptoms in most people, but are a leading infectious disease causing infant death and hospitalization and can result in increased morbidity and mortality in older adults and at-risk individuals. Pfizer has developed Abrysvo^®^, an unadjuvanted bivalent recombinant protein subunit vaccine containing prefusion-stabilized fusion (F) proteins representing RSV A and RSV B subgroups (RSVpreF). It is the only RSV vaccine approved for both maternal immunization to protect infants and active immunization of older adults (≥60 years) and 18–59-year-old individuals with high-risk conditions for prevention of RSV disease. **Methods**: Nonclinical safety studies, including a repeat-dose toxicity (RDT) study in rats and a combined developmental and reproductive toxicity (DART) study in rabbits, were conducted to support early clinical development. Study designs and parameters evaluated in these studies were consistent with principles and practices as outlined in relevant regulatory guidelines. RSVpreF bivalent vaccine, with or without Al(OH)_3_, was administered intramuscularly (IM) at 2× the human dose to animals in both studies. **Results**: Locally tolerated, reversible, inflammatory responses at the injection sites and the draining lymph nodes were observed as typical findings following vaccination. No effect of RSVpreF, with or without Al(OH)_3_, was observed on female fertility or on embryo–fetal or postnatal survival, growth, or development in the DART study. In both studies, robust immune responses to both RSV A and B antigens were observed, especially with the Al(OH)_3_ formulation. **Conclusions**: RSVpreF was well-tolerated both locally and systemically without any adverse effects on reproductive and developmental endpoints.

## 1. Introduction

Respiratory syncytial virus (RSV) is the most important cause of lower respiratory tract disease (LRTD) and a leading infectious disease causing death in young children, especially during their first 6 months of life and in low and middle income countries (LMICs) [1,2]. RSV can also cause repeat infections throughout life but poses particular risks to people with compromised immune, pulmonary, and/or cardiovascular functions and other chronic medical conditions [3,4,5,6]. RSV disease in older adults is associated with increased morbidity and mortality, either caused by the virus itself, due to bacterial superinfection, or deterioration of existing chronic medical conditions [7]. It has been estimated that, each year in the United States, RSV infections cause ~80,000 hospitalizations among infants [8] and 100,000~160,000 hospitalizations among adults aged ≥65 years [9,10]. Globally, while RSV-associated morbidity is high across nations with different income levels, mortality rate is particularly high in LMICs, as indicated by a systematic analysis on RSV disease burden datasets for children aged 0–60 months, showing that more than 97% of RSV-attributable deaths across all age bands were in LMICs in 2019 [1].

Despite the fact that RSV was discovered more than 65 years ago [11,12], developing a safe and effective RSV vaccine has proven very challenging. Decades of preclinical and clinical research have not only deepened scientific understanding of RSV pathogenesis but narrowed the focus on the prefusion, rather than post-fusion, state of F protein, a class I fusion glycoprotein of RSV, as a primary RSV antigen target for vaccine development [6,13,14,15,16]. These efforts led to market approvals of three vaccine products (prefusion F protein–based Arexvy^®^ from GSK and Abrysvo^®^ from Pfizer, and prefusion F mRNA-based mResvia^®^ vaccine from Moderna) in the past year (May 2023–May 2024) for immunization in older adults.

In addition to the older adult and high-risk adult indications, Abrysvo^®^ is also approved for protection of infants from birth through 6 months of age via maternal immunization. In the 1960s, observations of vaccine-mediated enhanced respiratory disease with a formalin-inactivated whole virus vaccine, a direct immunization approach aiming to protect infants during the peak RSV disease period (1–2 months of age), created barriers to vaccine development in RSV naive children. As adults are universally RSV-experienced [17], maternal immunization overcomes these barriers, relying on vaccination of pregnant women whose antibodies are trans-placentally transferred in utero to provide the newborn with immediate protection at birth from potentially life-threatening RSV infections. Abrysvo^®^ was successfully developed as the first and only vaccine given to pregnant women to prevent LRTD caused by RSV in babies from birth through 6 months of age, a critical period for RSV susceptibility [18,19,20,21].

Abrysvo^®^ is an unadjuvanted bivalent recombinant protein subunit vaccine containing prefusion-stabilized fusion (F) proteins representing RSV A and RSV B subgroups (RSVpreF). During the lead optimization stage, hundreds of engineered F constructs were evaluated in vitro and in vivo, resulting in the selection of lead construct 847 for introduction onto F glycoprotein backbones of RSV strains, representing the dominant circulating genotypes of the two major subgroups, A and B [15]. Here, we report the results of nonclinical safety studies conducted on candidate formulations (RSVpreF bivalent vaccine, containing both 847A and 847B antigens), including a repeat dose toxicity (RDT) study in rats and a combined developmental and reproductive toxicity (DART) study in rabbits, to support the clinical development and eventual licensure of the vaccine candidate for adult and maternal indications. The objective of the RDT study was to evaluate potential local and/or systemic toxicities of the RSVpreF bivalent vaccine, with or without Al(OH)_3_, after three separate IM injections. The DART study evaluated the effects of RSVpreF bivalent vaccine, with or without Al(OH)_3_, on female fertility, pregnancy, embryo–fetal development (including assessment of developmental toxicity), parturition and lactation, as well as the postnatal growth and development of offspring, to support clinical administration in women of child-bearing potential (WOCBP) and maternal immunization.

## 2. Materials and Methods

### 2.1. Animals and Husbandry

All animal care and experimental procedures were conducted in compliance with guidelines for the care and use of laboratory animals [22] as well as local regulations, and were reviewed and approved by an Institutional Animal Care and Use Committee (RDT study; AUP-GTN-2011-00314) or ethical committee (DART study, authorization no. 2017112118149040). Each study was conducted according to GLP and OECD guidelines.

Male and female Wistar Han rats were supplied by Charles River Laboratories (CRL), Raleigh, NC (10 weeks old at dose initiation) for the RDT study. For the DART study, male (20–23 weeks old at the start of mating) and virgin female (16–18 weeks old at dose initiation) New Zealand White (NZW) rabbits were supplied by CRL France, Chatillon sur Chalaronne, France. The selection of rats and rabbits as the toxicology test species and the overall study designs are consistent with relevant regulatory guidelines [23,24,25,26]. Rats and rabbits were both considered immunologically relevant species because both demonstrated functional immune responses to both A and B antigens. However, as shown in the RDT study results, the immune response to RSVpreF vaccine without Al(OH)_3_ in female rats was poor, which led to the selection of rabbits as the test species for the DART study.

Animals were offered food (Certified Irradiated Rodent Diet 5002 [PMI Feeds Inc., Richmond, IN, USA] for RDT; Pelleted Complete rabbit diet [3409 CRL, Kliba Nafag, Kaiseraugst, Switzerland] for DART) and locally sourced water ad libitum, except when fasting was conducted prior to clinical pathology collections or euthanasia. Animals were housed individually throughout the study in polycarbonate cages with ALPHA-dri^®^ bedding (Shepherd Specialty Papers Inc., Watertown, TN, USA) for the RDT study. For the DART study, rabbits were housed individually in composite plastic and metal cages compliant with European Regulations (Directive 2010/63/EU), except when each female was temporarily placed in the cage of the male for mating. Environmental conditions across studies were set to maintain relative humidity at 30–70% for rats and >35% for rabbits, and temperature at 20–26 °C for rats and 17–23 °C for rabbits, with room lighting set to provide a 12-h light/dark cycle.

### 2.2. Test and Control Articles

The test articles are formulations of the RSVpreF bivalent vaccine, with or without 0.4 mg/mL aluminum in Al(OH)_3_. Each 0.5 mL dose of the RSVpreF bivalent vaccine (used interchangeably hereafter as “RSVpreF”) contained 120 μg of 847A and 120 μg of 847B [15] for a total of 240 μg antigens, provided as a lyophilized dosage form by Pfizer Vaccines Research and Development. For both studies, 0.9% sodium chloride (sterile saline) and Al(OH)_3_ vehicle formulation were used as controls. The 240 μg total antigen dose was selected to support the highest anticipated human dose in clinical development and, therefore, represented 2× selected human dose for the commercial product, Abrysvo^®^.

On the day of dosing, the lyophilized RSVpreF dosage form was reconstituted using diluents provided by Pfizer Vaccines Research and Development so that the final dose formulations would contain identical compositions to those in the vehicle control buffer, except that the RSVpreF alone formulation was without Al(OH)_3_. The final vaccine formulation buffer for the RDT study contained 20 mM Tris, 50 mM NaCl, 0.02% polysorbate 80, and 9% sucrose, pH 7.4. For the DART study, the final vaccine formulation buffer contained 15 mM Tris, 37.5 mM NaCl, 0.015% polysorbate 80, 2.25% sucrose, and 4.5% mannitol, pH 7.4. The slight differences in the buffers reflected formulation development in support of different stages of clinical trials. These differences were not considered to have impacted the nonclinical safety assessment of the vaccine, as all the components are inactive ingredients commonly used in injectables and are within the ranges used in commercial vaccines; in addition, equivalent RSVpreF formulations had been evaluated clinically.

### 2.3. Study Design

#### 2.3.1. 38-Day Repeat-Dose Toxicity Study in Rats with 1-Month Recovery

Wistar Han rats were randomized into 4 groups (15/sex/group) based on their body weights: (1) Saline Control; (2) Al(OH)_3_ Vehicle Control; (3) RSVpreF; (4) RSVpreF with Al(OH)_3_. Doses were administered by two separate intramuscular injections (0.25 mL per site) on each dosing day (Days 1, 22, and 36, reflecting the N + 1 strategy for vaccine administration [25,26]) into each of the left and right quadriceps muscles of an animal in the respective group, for a total of 0.5 mL (the human dose volume). On the day after the dosing phase necropsy day (Day 38), a 4-week recovery phase started (Figure 1A).

Clinical observations were performed daily. Body weights and food consumption were measured at least weekly. Body temperatures were measured once prior to the initiation of dosing, pre-dose on Dosing Phase Days 1, 22, and 36 and at approximately 4 and 24 h post-dose for all animals. Injection site scoring (dermal assessment utilizing the modified Draize Method [27]) was assessed on dosing days before dosing and 4 and 24 h post-dose. Blood samples were taken on Days 3, 22 and 38 of the dosing phase and at the end of the recovery phase for clinical laboratory measurements. Serology samples were taken prior to initiation of dosing, at the end of the dosing phase (Days 36 and 38), and during the recovery phase (Days 8/7 and 28/27, for males/females).

Animals were fasted overnight and euthanized on dosing phase Day 38 (2 days after the last dosing day, the first 10 animals/sex/group) or on the last day of the recovery phase (remaining 5/sex/group). Necropsy, tissue collection, organ weights, macroscopic tissue evaluation, and microscopic examination were performed on all animals according to the WHO guidelines for nonclinical evaluation of vaccines and adjuvanted vaccines [25,26]. Necropsy included a macroscopic examination of the external surface of the body, the thoracic and abdominal cavities and their contents, and the collection of all major tissues and macroscopic findings. Adrenal, brain, epididymis, heart, kidney, liver, ovary, prostate, spleen, testis, and thymus weights were collected. Organ-to-body weight and organ-to-brain weight ratios were calculated. All tissues collected for histopathological evaluation were embedded in paraffin wax, sectioned, and stained with hematoxylin and eosin prior to microscopic examination.

#### 2.3.2. Developmental and Reproductive Toxicity Study in Rabbits

The DART study was conducted by Charles River Laboratories France Safety Assessment SAS. Virgin female Crl:KBL(NZW) rabbits (approximately 16–18 weeks old at first dose and 3.1–4.6 kg at mating) were acclimated and randomly assigned to groups in two main cohorts (n = 22 per dose group in the Caesarean Cohort and n = 22 per dose group in the Littering Cohort). Female rabbits were administered saline, Al(OH)_3_ vehicle, RSVpreF, or RSVpreF with Al(OH)_3_ 21 and 7 days prior to the start of mating with non-dosed males and on gestation days (GD) 10 and 24, for a total of 4 doses. Each dose consisted of one 0.5 mL intramuscular injection to alternating vastus lateralis (quadriceps) muscle, starting with the left. An overview of the study design is depicted in Figure 1B, and the study was designed for compliance with regulatory guidance at the time, including the number of animals per group [23,24].

Clinical signs, body weight, and food consumption were monitored throughout the study. Female rabbits were paired (1:1 ratio) with non-dosed males of the same strain for at least 3 min or until copulation was observed, for a maximum of 5 days. The day of mating was considered as GD 0.

For animals in the Caesarean Cohort, scheduled necropsy was performed on animals euthanized by sodium pentobarbitone injection followed by exsanguination on GD 29. Following a gross examination of the abdominal, thoracic, and pelvic viscera, the ovaries and uterus were removed and examined, including examination of the placentae. The following data were recorded: pregnancy status, numbers of corpora lutea and implantations, number and distribution of live fetuses and embryo/fetal deaths, gravid uterus weight, and individual fetal weights. The uteri of apparently non-gravid females were placed in ammonium sulfide solution in order to stain any previously undetected implantation sites. All live fetuses were examined for external defects and euthanized by an oral administration of sodium pentobarbitone. Fetuses were then examined viscerally and sexed at the time of Caesarean section. Following this, the head of approximately half of the fetuses in each litter was removed and fixed in Harrison’s fluid for subsequent examination by serial sectioning. The eviscerated fetal carcasses were fixed and processed for skeletal examination. The skeletal examinations were performed following maceration of the soft tissues with aqueous potassium hydroxide, staining of the skeleton with Alizarin red then passage into glycerol. Soft tissue and skeletal examination were performed using a binocular microscope.

For animals in the Littering Cohort, from GD 30, each female was observed at least 4 times a day for the onset and duration of parturition. Date of parturition (lactation day [LD] 0 or postnatal day [PND] 0), duration of gestation (calculated value based on the time of onset of parturition), abnormalities of nesting or nursing behavior, and number of implantation sites (at necropsy) were recorded. For each litter, external abnormalities of the kits, number and weight of kits alive on various PNDs, pre-weaning development of the offspring as assessed by verification of pupil and auditory reflexes on PND 35 were recorded. On LD 35, the scheduled necropsy was performed on animals euthanized by sodium pentobarbitone injection followed by exsanguination. Following a gross examination of the abdominal, thoracic, and pelvic viscera, the ovaries and uteri were examined for any abnormalities. The number of former implantation sites in the uteri were also recorded. All surviving kits in the Littering Cohort were euthanized by intramuscular injection of ketamine–xylazine followed by cardiac injection of sodium pentobarbitone. All kits (including any moribund or found dead) were sexed by internal inspection and were macroscopically examined (including the thoracic, abdominal, and pelvic viscera) for structural or pathological changes.

### 2.4. Immunogenicity Evaluation

In the RDT study, approximately 0.7–1.0 mL of blood was collected from each animal into serum separator tubes prior to initiation of dosing, on dosing phase Day 36 and Day 38, and on recovery phase Days 8/7 (males/females) and 28/27 (males/females). In the DART study, approximately 2 mL blood was collected in serum separator tubes on all surviving females 21 and 1 days before mating, and on GD 29 for Caesarean Cohort females and LD 35 for the Littering Cohort animals just before necropsy. On GD 29, fetal blood was collected and pooled from at least 4 fetuses from each Caesarean Cohort litter, where possible. On PND35, blood was collected and pooled from at least 4 viable kits per Littering Cohort litter, where possible. See Figure 1 for sampling within the context of the overall study design.

After collection, the blood samples were centrifugated at 1800× *g*, at approximately +4 °C for 10 min within 1 h after sampling. The serum was separated, aliquoted, and stored at −20 °C or lower before shipping to Pfizer Vaccine Research and Development for analysis.

Details of the RSV Microneutralization Assay (RSV-NT) and functional antibody measurements were described previously [28]. Briefly, serial dilutions of heat-inactivated sera were mixed with RSV subgroup A or B strain separately before being transferred onto an A549 epithelial cell monolayer in a 384-well tissue culture plate. After 22 to 26 h, viral foci were detected using a mouse anti-RSV fusion glycoprotein (F) monoclonal antibody followed by an Alexa488-conjugated anti-mouse secondary antibody stain. Fluorescently labeled viral foci were enumerated and neutralization titers were calculated as the interpolated reciprocal of the serum dilution, resulting in 50% reduction in the number of viral focus forming units when compared to the control without test serum.

Assay results of the functional antibody responses were Log10 transformed, and the average Log10 neutralization titers and 95% confidence intervals (CI) were calculated for each of the study groups at each time point. The data were then back transformed to generate GMT and the corresponding 95% CI on the arithmetic scale. The pairwise geometric mean ratio (GMR) and 95% CI were calculated for each study group.

### 2.5. Statistical Analysis

Descriptive statistics were generated for each parameter and group at each scheduled sampling time or each time interval. Statistical tests were conducted at the 5% and 1% significance levels.

#### 2.5.1. Repeat-Dose Toxicity Study

Statistical analyses of body weight, body weight change, food consumption, body temperature, injection site scores, and organ weight data were conducted by the Statistics Department using in-house developed iStats v1.0 with the methods outlined below. All analyses were performed separately for each sex.

Analyses of body weight and food consumption parameters were carried out on measurements collected for each animal at the scheduled sampling times or time intervals. In addition, body weight change at selected intervals was analyzed. Analysis of body temperature was based on the maximum body temperature post injection for each animal. Analysis of injection site score was based on the average irritation score post injection for each animal. Analyses of organ weight parameters were performed on measurements collected for each animal at the scheduled sampling times or time intervals. In addition, organ weight to body weight and organ weight to brain weight ratios were analyzed.

A nonparametric (rank-transform) one way analysis of variance (ANOVA) was conducted, with a two-sided pairwise comparisons of Groups 1 and 4 to Group 2 and Group 3 to Group 1 using Tukey’s test. Average ranks were assigned to ties.

#### 2.5.2. Developmental and Reproductive Toxicity Study

Combined maternal body weight, body weight gain and food consumption data for F0 dams from the Caesarean and littering subgroups up to and including the gestation phase and F1 offspring functional tests were analyzed using the SAS software package (Version 2.00). Levene’s test was used to test the equality of variance across groups and Shapiro–Wilk’s test was used to assess the normality of the data distribution in each group. Data with homogenous variances and normal distribution in all groups were analyzed using ANOVA followed by Dunnett’s test. Data showing non-homogenous variances or a non-normal distribution in at least one group were analyzed using the Kruskal–Wallis test followed by Wilcoxon’s rank sum test.

Statistical analyses of all other endpoints were performed in the Provantis data acquisition system. The best transformation for the data (none, log or rank) was determined depending upon the normality of the data distribution tested by the Shapiro–Wilk test and the homogeneity of the variances across groups tested by the Bartlett’s test. Non- or log-transformed data were analyzed by parametric methods. Rank transformed data were analyzed using non-parametric methods. The data from each test item group and the vehicle control group were analyzed by parametric or non-parametric Dunnett’s test to look for significant differences from the saline control group. The number of resorptions, number of dead fetuses and all litter-based percentages were analyzed using non-parametric methods, i.e., Kruskal–Wallis test, followed by non-parametric Dunnett’s test if the Kruskal–Wallis was significant. Selected incidence data were analyzed using a chi-squared test for all groups followed by Fisher’s two-tailed test with Bonferroni correction for each treated group versus the control if the chi-squared was significant.

## 3. Results

### 3.1. 38-Day Repeat-Dose Toxicity Study in Rats with 1-Month Recovery

#### 3.1.1. In-Life Observations

IM administration of RSVpreF, with or without Al(OH)_3_, on Days 1, 22, and 36 was tolerated, with no clinical signs related to the vaccine administration. All animals survived to the scheduled termination day at the end of the dosing or recovery phase of the study.

There were no effects on the mean body weight (Figure 2) or mean food consumption (Appendix A) during the dosing or recovery phase of the study associated with administration of RSVpreF with or without Al(OH)_3_ compared with the Al(OH)_3_ control group or saline control group, respectively. There were also no differences noted in these parameters when the Al(OH)_3_ control group was compared with the saline control group.

No increase in mean maximum body temperature was observed in animals administered RSVpreF, with or without Al(OH)_3_, compared with the Al(OH)_3_ control or saline control, respectively. Statistically significant higher mean maximum body temperatures were noted in males administered RSVpreF with Al(OH)_3_ on Days 1 and 22 compared with the Al(OH)_3_ control and in males administered RSVpreF on Day 22, as compared with the saline control (Table 1). However, these slight differences (<1 °C) were not considered related to the test article because individual values fell within the background range observed prior to initiation of dosing or pre-dose on both days without a clear pattern of post-dose temperature increase.

There were no effects at the injection sites associated with administration of RSVpreF, with or without Al(OH)_3_, compared with the Al(OH)_3_ control group or saline control group, respectively. The dermal scores in all groups were grade 0 (no erythema or edema) for erythema or edema on dosing phase Days 1, 15 and 29.

No ophthalmologic findings were associated with administration of RSVpreF, with or without Al(OH)_3_, compared with the Al(OH)_3_ control group or saline control group, respectively, at the end of the dosing phase.

#### 3.1.2. Clinical Laboratory Measurements

There were no effects on hematology, coagulation, clinical chemistry, or urinalysis parameters (Appendix A) associated with administration of RSVpreF without Al(OH)_3_ compared to the saline control group.

RSVpreF with Al(OH)_3_-related higher mean neutrophil counts and fibrinogen values were observed at the end of the dosing phase compared with the Al(OH)_3_ control group, and fully recovered. In addition, higher mean globulin and alpha-2-macroglobulin (α2-AM) values and lower mean albumin and albumin/globulin (A/G) ratios were observed at the end of the dosing phase compared with the Al(OH)_3_ control group (Figure 3). The effect on mean globulin and AG ratio were still present at the end of the recovery phase, while the effect on mean albumin and α2-AM (Figure 3) was fully recovered. There were no effects on urinalysis parameters associated with administration of RSVpreF with Al(OH)_3_ compared with the Al(OH)_3_ control group (Appendix A).

#### 3.1.3. Antibody Response to Vaccine Components

As shown in Figure 4, without Al(OH)_3_, the RSVpreF bivalent vaccine formulation elicited measurable RSV A and RSV B neutralizing antibody responses in males and limited to little or no response in females compared with an overall higher and consistent response in males and females with the RSVpreF with Al(OH)_3_ formulation. RSVpreF with Al(OH)_3_ responses in both sexes and RSVpreF responses in most males remained at the end of the recovery phase. No antibody responses were observed in the animals administered saline (Group 1) or in Al(OH)_3_ vehicle controls (Group 2).

#### 3.1.4. Postmortem Observations

There were no effects on mean absolute or relative (to body weight or brain weight) organ weights (Appendix A) associated with administration of RSVpreF with or without Al(OH)_3_ compared with the Al(OH)_3_ vehicle control group or the saline control group, respectively, at the end of the dosing and recovery phase of study.

Enlargement of the draining lymph nodes was identified at the end of the dosing phase necropsy in males and females administered RSVpreF with or without Al(OH)_3_, and fully recovered. This finding was also present in one female administered the Al(OH)_3_ vehicle control at the end of the dosing phase, but was not present in any of the saline control animals. Enlarged draining lymph nodes correlated microscopically with minimal to mild increased cellularity of germinal centers.

At the end of the dosing phase, chronic active inflammation was present at the injection site in all groups; however, the severity was greater in animals administered RSVpreF with Al(OH)_3_ or the Al(OH)_3_ vehicle control (Figure 5). In the draining lymph nodes, there was increased cellularity in germinal centers in all groups; however, it was generally present at a higher incidence in animals administered RSVpreF with or without Al(OH)_3_ (Figure 6). Additionally, there was accumulation of macrophages in the draining lymph nodes in animals administered RSVpreF with Al(OH)_3_ or the Al(OH)_3_ vehicle control.

At the end of the 28-day recovery phase, microscopic findings at the injection site fully recovered in animals administered RSVpreF without Al(OH)_3_ or saline, and partially recovered in animals administered RSVpreF with Al(OH)_3_ and did not recover in animals administered only the Al(OH)_3_ vehicle control. In the draining lymph nodes, a full recovery of increased cellularity in germinal centers was observed in males administered RSVpreF with Al(OH)_3_ and in animals administered RSVpreF without Al(OH)_3_ or the Al(OH)_3_ vehicle control, and a partial recovery was observed in females administered RSVpreF with Al(OH)_3_. However, accumulation of macrophages in the draining lymph node did not recover in the animals administered RSVpreF with Al(OH)_3_ or the Al(OH)_3_ vehicle control.

### 3.2. Developmental and Reproductive Toxicity Study in Rabbits

No mortality related to RSVpreF, with or without Al(OH)_3_, was observed in the study. Two females aborted in late gestation (1 in the Al(OH)_3_ control group on GD 26 and 1 in the RSVpreF with Al(OH)_3_ group on GD 24). In the absence of any noteworthy macroscopic findings at necropsy, any RSVpreF-related increase in perinatal death, and the presence of abortions in historical control data, these isolated events were not considered vaccine-related. All kits from 2 litters died 4–5 days after delivery (1 in the Al(OH)_3_ control group on LD 5 and 1 in the RSVpreF with Al(OH)_3_ group on LD 4). In the absence of any noteworthy macroscopic findings in females or kits and the absence of any related observations in other animals (postnatal survival), these isolated events were also not considered vaccine-related.

There were no RSVpreF (with or without Al[OH]_3_) related clinical signs (including at the injection sites), or changes in the mean body weight or body weight gain, or changes in the mean food consumption observed during the premating, gestation, or lactation periods (Appendix A).

#### 3.2.1. Assessment of Female Fertility and Pregnancy

There was no RSVpreF-related effect, with or without Al(OH)_3_, on mating performance or fertility (Table 2).

In total (Caesarean and littering cohorts combined), 43, 41, 40, and 40 (out of 44) females were mated in the saline control, Al(OH)_3_ vehicle control, RSVpreF, and RSVpreF with Al(OH)_3_ groups, respectively, after completion of the 5 days of pairing. The copulation index for the RSVpreF groups, with and without Al(OH)_3_ (91%), was marginally lower than in the saline and control and Al(OH)_3_ vehicle control groups (98% and 93%, respectively) but remained within the historical control range (89% to 100%), and thus was considered as unrelated to RSVpreF administration, with and without Al(OH)_3_. In total, there were 41/43, 40/41, 39/40, and 38/40 pregnant/mated females in the saline control, Al(OH)_3_ vehicle control, RSVpreF, and RSVpreF with Al(OH)_3_ groups, respectively, resulting in a fertility index that was 95% to 98% across groups. The pregnancy rate (number pregnant/number paired × 100) was 93%, 91%, 89%, and 86% in the saline control, Al(OH)_3_ vehicle control, RSVpreF, and RSVpreF with Al(OH)_3_ groups, respectively; all within the historical control range of 75% to 95.5%.

#### 3.2.2. Caesarean Section Cohort

There were no RSVpreF-related effects, with or without Al(OH)_3_, on any Caesarean section parameters (Table 2). The mean numbers of corpora lutea and implantation sites in the RSVpreF with Al(OH)_3_ group (9.1 and 8.2, respectively) were slightly lower, and the percentage pre-implantation loss slightly higher (10.9%) compared with the saline control group (10.1, 9.5, and 5.5%, respectively). However, as these values were not statistically significant and were within the historical control range for these endpoints (and similar to the historical control means for each), none of these numerical differences were considered as vaccine-related. This is further supported by the lack of any comparable finding for the prebirth data in the littering subgroup, including number of implantations and percentage of pre-birth loss. There was no RSVpreF-related effect, with or without Al(OH)_3_, on mean fetal weight or fetal sex ratio.

There was no RSVpreF-related effect, with or without Al(OH)_3_, on fetal external, visceral, or skeletal morphology (Appendix A). Sporadic abnormalities observed were not considered related to RSVpreF (with or without Al(OH)_3_ because they were observed in isolation (single incidence), at a similar incidence to the saline and/or the Al(OH)_3_ vehicle control groups, and/or were within the Test Facility historical control data. There were two fetuses from different litters in the RSVpreF without Al(OH)_3_ group that had multiple abnormalities of the thoracic vertebrate in the absence of occurrence in control fetuses; however, this find was not considered vaccine-related because it had been observed numerous times in the Test Facility historical control data and it was not observed in the RSVpreF with Al(OH)_3_ group.

#### 3.2.3. Delivery Cohort

There were no RSVpreF-related effects, with or without Al(OH)_3_, on parturition or gestation length (all females had liveborn kits and mean gestation length was comparable) in all groups. No RSVpreF-related effects, with or without Al(OH)_3_ were observed in the delivery cohort (Table 2). The mean numbers of implantation sites and number of live kits born per litter in the vaccine groups were comparable with, or superior to, that in the saline control group. Although pup survival appeared slightly lower in the Al(OH)_3_ control group (lactation index) and the RSVpreF with Al(OH)_3_ group (viability index), one female in each group had total litter death on PND 5 and PND 4, respectively, that accounted for most of that (Table 2, footnote f). As total litter death post-partum had also been observed in the historical control data, the presence of these isolated findings in each of these two groups was considered incidental. In addition, both the viability and lactation indices in the vaccine groups were consistent with the saline control and within the historical control ranges. There was no pattern in the incidence of pup clinical observations that suggested any RSVpreF-related effect, with or without Al(OH)_3,_ and no effect on the mean pup weight was observed. No RSVpreF-related effect, with or without Al(OH)_3_, on the pre-weaning functional reflexes (pupil and auditory reflexes) was observed (Appendix A). No pattern in the incidence of pup macroscopic observations suggested any RSVpreF-related effect, with or without Al(OH)_3_.

#### 3.2.4. Immunogenicity Evaluation

As shown in Figure 7, the neutralization titers demonstrated that the animals administered RSVpreF, both with and without Al(OH)_3_, elicited an immune response to RSV A and RSV B, and these responses were detectable in fetuses and kits from the Caesarean and Littering Cohorts, respectively. The animals in the saline and Al(OH)_3_ control groups did not elicit an immune response to RSV A or RSV B.

## 4. Discussion

Bivalent RSVpreF, with or without Al(OH)_3_, was evaluated in a RDT study in rats and in a combined DART study in rabbits. In both studies, the vaccine doses were 240 μg total antigens (120 μg each for A and B) to support the anticipated highest clinical dose. Subsequent clinical trial data demonstrated that 120 μg total antigen (60 μg each for A and B) [18,20,29,30] without Al(OH)_3_ was safe and effective, which was the selected dose and formulation of Abrysvo^®^ for both maternal administration and for older adults. Therefore, the nonclinical safety evaluation was carried out at 2× the commercial dose, which is higher than the WHO guideline recommendation [25,26], but in alignment with the clinical development plans. On a body weight basis, the nonclinical dose represents approximately 40× (rabbit) or 480× (rat) the human dose (2 μg/kg, assuming an average of 60 kg body weight).

For the clinically relevant formulation, RSVpreF without Al(OH)_3_, the only findings in the rat RDT study were non-adverse chronic active inflammation at the injection site and higher cellularity in the germinal center (correlating with macroscopic enlargement) of the draining lymph nodes, with both fully recovered at the end of the 1-month recovery phase. With Al(OH)_3_ in the formulation, more profound (higher incidence and/or severity) microscopic findings at the injection sites and the draining lymph nodes were observed, along with neutrophil and fibrinogen increases and clinical chemistry changes (higher globulin and alpha-2-macroglobulin and lower albumin and A/G ratio), indicative of an inflammatory response. All of these changes were consistent with findings typically observed with IM administration of vaccines (especially aluminum-containing vaccines) [31,32,33,34] and were reflective of innate immune responses essential for the development of strong and long-term vaccine efficacy and thus are considered desired and expected findings [35,36,37].

Despite robust RSVpreF responses in rabbits in the DART study, a weak neutralizing antibody response to both RSV A and B antigens was elicited in the male rats in the RDT by the RSVpreF vaccine without Al(OH)_3_ and only a small number of female animals had higher antibody titers to RSV A antigen. For this reason, rabbits were selected for evaluation of the vaccine candidates in the combined DART study. Nonetheless, RSVpreF with Al(OH)_3_ elicited a robust neutralizing antibody response, providing an opportunity to evaluate the toxicity (if any) of the antigen-specific immune response. Interestingly, a robust immune response to both RSV A and B antigens was observed in female rabbits as well as in humans, including non-pregnant and pregnant women [19,38,39], by the RSVpreF bivalent vaccine without any adjuvant, indicating that the lack of antibody responses in female rats may be a species-specific observation.

In addition to the robust immune response in female rabbits in the DART study, the neutralizing antibodies to both RSV A and B antigens were also present in rabbit fetuses at the end of gestation (GD 29) and kits at the end of lactation (PND 35). No effect of the RSVpreF bivalent vaccine (despite being evaluated at higher than the clinical dose), with or without Al(OH)_3_, was observed on female fertility, or on embryo–fetal or postnatal survival, growth, or development in the rabbit DART study. This lack of findings in the RSVpreF rabbit DART study is consistent with the lack of adverse effects observed in the rat and rabbit DART studies conducted to support the GSK RSVPreF3 vaccine [40]. None of the results of these nonclinical DART studies would predict an increased risk of preterm birth in humans; however, the development of RSVpreF3 vaccine for the maternal immunization indication was halted in February 2022 due to an unexpected increased risk of preterm birth and neonatal deaths in the vaccine group as compared with the placebo group, mainly seen in low- and middle-income countries [41]. On the other hand, Abrysvo^®^ (RSVpreF) [18,20,21,38,42] is approved for use during pregnancy to prevent LRTD caused by RSV in babies from birth through 6 months of age. Although a numerical imbalance in preterm births in RSVpreF recipients was observed in two clinical studies, the differences were not statistically significant and neonatal deaths were lower in the vaccine group than the placebo group [43]. In a recent Advisory Committee on Immunization Practices (ACIP) meeting, Dr. Pedro Moro from the US Centers for Disease Control and Prevention (CDC) presented post-licensure safety surveillance data of Abrysvo^®^ in pregnant women during the 2023–2024 season, which showed a 4.1% incidence of preterm births among pregnant women who received the vaccine [44]. This incidence rate is within the expected range of 3.1–6.1% incidence rate of preterm births among women at 32–36 weeks gestation prior to the introduction of the vaccine [44].

*Limitations of studies*: A number of limitations in nonclinical safety studies for vaccine evaluation have been acknowledged in regulatory guidance and literature [26,45,46,47]. For instance, physiological differences in system biology (especially in the immune system and reproductive and developmental biology) can result in different responses among species to a vaccine antigen, adjuvant, and/or other components. Similarly, local and systemic effects observed in a nonclinical safety study may not be directly translatable to the clinic, and effects observed in homogeneous healthy animals may not represent those in heterogenous human population, especially when accompanied by various diseases/conditions. Certain types of reactogenicity effects that are commonly observed in human trials (e.g., headache, fatigue) may be impractical to detect in animals or vice versa (e.g., microscopic findings in animals). Due to size differences between nonclinical species and humans (e.g., ~0.2 kg rats vs. ~50 kg human adults:), there may be an exaggerated effect of the vaccine when being administered at a full (in our case, 2×) human dose to an animal. Additionally, a rare and/or late onset adverse effect that may occur in humans may not be observed in animal studies. For the DART study, the evaluation of potential developmental effects was based on the periodic dosing during gestation which may not necessarily include peak exposure during all prenatal and postnatal developmental windows of all developing organ systems in offspring.

Despite these limitations, nonclinical studies remain a suitable approach to assessing the safety of a vaccine candidate and a prerequisite for entering human trials. Indeed, our data enabled the clinical development of the RSVpreF vaccine in various populations, which demonstrated a favorable benefit–risk profile that led to market authorization approval for both maternal and active immunization indications.

## 5. Conclusions

The nonclinical safety profile of RSVpreF, with or without Al(OH)_3_, demonstrated a tolerable nonclinical safety profile supporting clinical development and licensure/marketing authorization.

## Figures and Tables

**Figure 1 vaccines-13-00026-f001:**
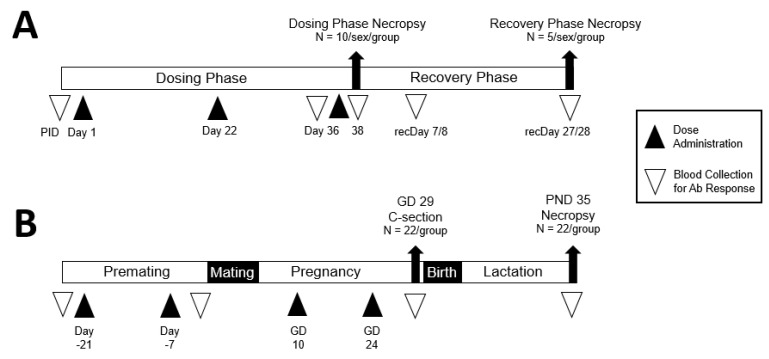
Overview of the study designs. (**A**) *RDT study in rats*: 4 groups of animals (15/sex/group) were respectively administered saline, Al(OH)_3_ vehicle control, RSVpreF, or RSVpreF with Al(OH)_3_ by IM injections on Days 1, 22, and 36. On Day 38, 10 animals/sex/group were necropsied after overnight fasting, and the remaining 5/sex/group entered the 4-week recovery phase. Blood collections were carried out prior to initiation of dosing (PID), on Days 36 and 38 of the dosing phase, and on Days (recDays) 7/8 and 27/28 (females/males) of the recovery phase for serology evaluation. (**B**) *DART study in rabbits*: Female rabbits (44/group) were administered four 0.5 mL IM injections (2 prior to cohabitation with the males and 2 during gestation) of saline, Al(OH)_3_ vehicle, RSVpreF, or RSVpreF with Al(OH)_3_. Half of each group (22/group) underwent Caesarean section on gestation day (GD) 29. The remaining rabbits (22/group) were allowed to deliver naturally, and the maternal animals and offspring were followed through to the end of weaning. Blood was collected for measurement of antibody response in maternal animals prior to the first dose, at mating, end of gestation (GD 29), and end of lactation (lactation day [LD] 35). Blood was collected from fetuses on GD 29 and from offspring at the end of lactation (postnatal day [PND] 35).

**Figure 2 vaccines-13-00026-f002:**
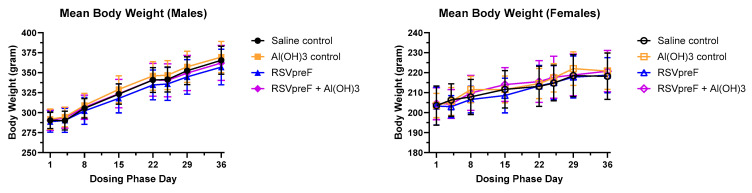
Effect of RSVpreF on the mean body weights of male and female rats. No statistically significant differences were observed on any day during the dosing and recovery phases between RSVpreF and saline control or between RSVpreF + Al(OH)_3_ and Al(OH)_3_ control.

**Figure 3 vaccines-13-00026-f003:**
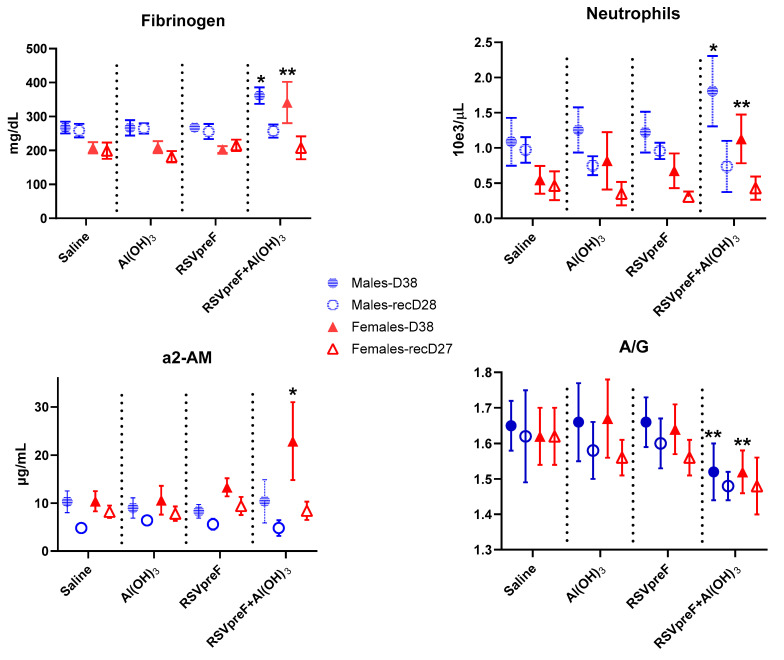
Effects of RSV-preF on fibrinogen, neutrophils, alpha-2-macroglobulin, and albumin/globulin ratio in male and female rats. Data from dosing Day 38 (D38) and recovery days (recD28 or recD27) are presented. Mean ± SD. * *p* < 0.05, ** *p* < 0.01 vs. saline or Al(OH)_3_ control at each time point, respectively, for RSVpreF or RSVpreF + Al(OH)_3_.

**Figure 4 vaccines-13-00026-f004:**
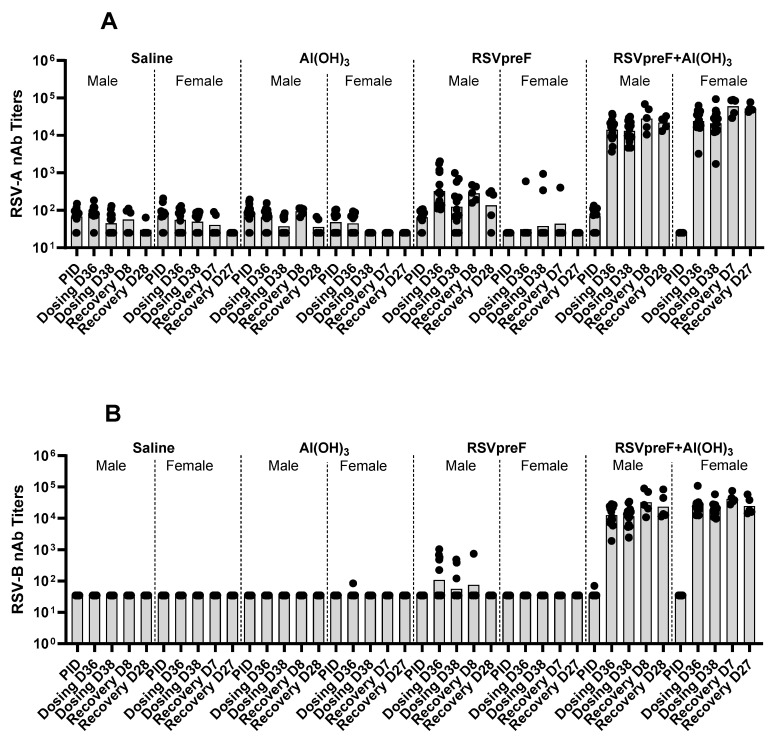
Neutralizing antibody responses to RSV A (**A**) and RSV B (**B**) antigens following the administration of RSVpreF, with or without Al(OH)_3_, in male and female rats. Serum samples were collected prior to initiation of dosing (PID), prior to dosing on Days 36 and 38 of the dosing phase, and on recovery phase Days 8/7 (males/females) and 28/27 (males/females). Individual (scattered) and geometric mean (bar) data are presented.

**Figure 5 vaccines-13-00026-f005:**
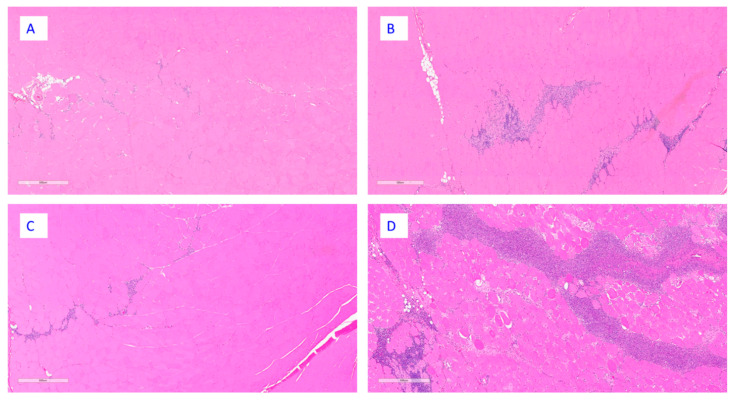
Histopathological features at the injection site from animals necropsied at the end of dosing phase. Representative images from animals administered saline (**A**), Al(OH)_3_ (**B**), RSVpreF (**C**), or RSVpreF + Al(OH)_3_ (**D**). The bar inside of each image depicts a scale of 500 µm; hematoxylin and eosin stain.

**Figure 6 vaccines-13-00026-f006:**
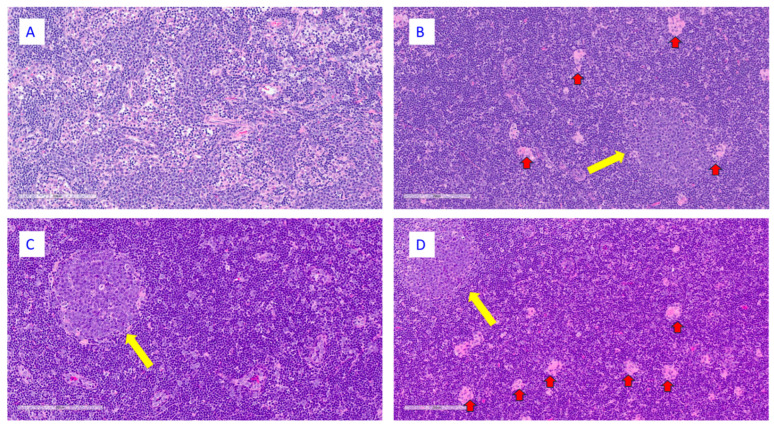
Histopathological features at the draining lymph node from animals necropsied at the end of dosing phase. Representative images from animals administered saline (**A**), Al(OH)_3_ (**B**), RSVpreF (**C**), or RSVpreF + Al(OH)_3_ (**D**). The bar inside of each image depicts a scale of 200 µm; hematoxylin and eosin stain. Yellow and red arrows indicate germinal centers and macrophages, respectively.

**Figure 7 vaccines-13-00026-f007:**
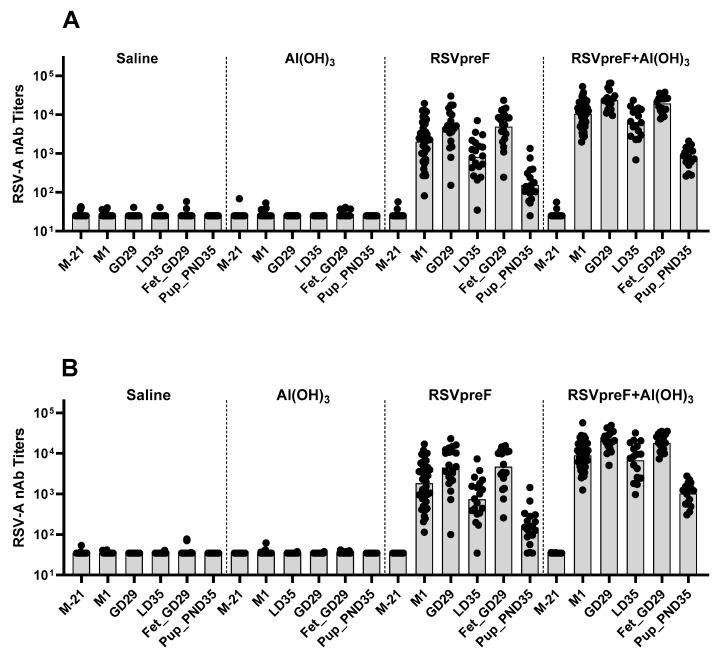
Neutralizing RSV-A (**A**) and RSV-B (**B**) antibody titers in animals administered RSVpreF, with or without Al(OH)_3_, as compared with those administered saline or Al(OH)_3_. Maternal blood samples were taken from all females of each group on premating day M-21 (21 days before mating, prior to the first injection), mating day 1 (M1, before mating), gestation day 29 (GD29, Caesarean Cohort) and lactation day 35 (LD35, Littering Cohort). Blood samples were also taken from surviving fetuses on GD 29 (Fet_GD29, Caesarean Cohort) and from all surviving kits on postnatal day 35 (Pup_PND35, Littering Cohort). Individual (scattered) and geometric mean (bar) data are presented.

**Table 1 vaccines-13-00026-t001:** Effect of RSVpreF on the mean maximum body temperature of male and female rats. Maximum post-dose body temperatures (either at 4 or 24 h post-dose) from each animal administered saline, Al(OH)_3_, RSVpreF, or RSVpreF + Al(OH)_3_ were analyzed and presented as group mean ± SD. * *p* < 0.05, ** *p* < 0.01 vs. saline or Al(OH)_3_ control, respectively, for RSVpreF or RSVpreF + Al(OH)_3_.

Dosing Day	Saline		Al(OH)_3_ Vehicle		RSVpreF		RSVpreF + Al(OH)_3_	
	M	F	M	F	M	F	M	F
Day 1	37.10 ± 0.51	37.92 ± 0.52	37.13 ± 0.65	37.81 ± 0.62	37.34 ± 0.49	38.21 ± 0.60	37.78 ± 0.49 *	38.29 ± 0.45
Day 22	36.32 ± 0.91	38.06 ± 0.45	36.41 ± 0.72	37.99 ± 0.76	37.09 ± 0.59 *	38.47 ± 0.54	37.30 ± 0.62 **	38.50 ± 0.32
Day 36	37.20 ± 0.52	38.43 ± 0.43	37.51 ± 0.67	38.27 ± 0.58	37.17 ± 0.79	38.38 ± 0.60	37.63 ± 0.62	38.61 ± 0.31

**Table 2 vaccines-13-00026-t002:** Fertility, C-section, and Delivery parameter results from female rabbits administered saline, vehicle, RSVpreF, or RSVpreF with Al(OH)_3_.

	Saline	Vehicle Al(OH)_3_	RSVpreF	RSVpreF + Al(OH)_3_	Historical Control Mean (Min–Max) ^a^
**Fertility (n)**	**44**	**44**	**44**	**44**	**-**
Copulation Index ^b,c^	43/44 (98%)	41/44 (93%)	40/44 (91%)	40/44 (91%)	95.1% (89–100%)
Fertility Index ^b,d^	41/43 (95%)	40/41 (98%)	39/40 (98%)	38/40 (95%)	93.7% (85–100%)
Pregnancy Rate ^b,e^	41/44 (93%)	40/44 (91%)	39/44 (89%)	38/44 (86%)	88.3% (75–95.5%)
**Caesarean Section Cohort (n)**	**20**	**18 ^f^**	**20**	**19**	**-**
Corpora lutea	10.1 ± 1.9	10.3 ± 1.9	9.5 ± 1.2	9.1 ± 1.9	10.8 (9.0–12.1)
Implantation sites	9.5 ± 2.1	9.7 ± 2.3	9.1 ± 1.4	8.2 ± 2.4	9.7 (8.0–10.9)
Pre-implantation loss (%)	5.5 ± 8.6	6.1 ± 13.9	3.9 ± 6.5	10.9 ± 15.5	9.9 (5.9–12.7)
Early Resorptions	0.2 ± 0.5	0.4 ± 0.7	0.2 ± 0.4	0.2 ± 0.5	0.3 (0.1–0.7)
Late Resorptions	0.2 ± 0.5	0.1 ± 0.2	0.2 ± 0.5	0.1 ± 0.2	0.3 (0.1–0.6)
Post-implantation loss (%)	3.5 ± 7.0	4.5 ± 7.2	3.9 ± 6.3	2.3 ± 6.0	5.8 (2.4–11.4)
Number live fetuses	9.2 ± 2.0	9.2 ± 2.3	8.8 ± 1.5	8.0 ± 2.4	9.1 (7.7–10.3)
Fetal body weight (g)	41.5 ± 4.2	43.4 ± 5.9	43.5 ± 4.1	43.0 ± 5.6	41.0 (38.4–43.2)
**Delivery Cohort (n)**	**21**	**21 ^g^**	**19**	**18 ^f,g^**	**-**
Gestation length	31.6 ± 0.7	31.2 ± 0.4	31.6 ± 0.6	31.3 ± 0.5	31.8 (31.3–31.8)
Former implantation sites	8.1 ± 2.3	9.8 ± 2.4	9.4 ± 2.0	8.4 ± 2.5	9.2 (8.4–10.5)
Kits born per litter	7.74 ± 2.0	9.5 ± 2.0	8.8 ± 2.1	8.1 ± 2.6	9.0 (8.0–9.8)
Viability index (%) ^h^	97.4	97.4	97.6	93.7	97.5 (91.1–100)
Lactation index (%) ^i^	95.4	89.5	93.8	93.2	90.2 (84.5–94.4)
Pup body weight on PND 4 (g)	89.6 ± 21.1	78.7 ± 13.0	83.4 ± 13.3	85.1 ± 14.6	85.4 (79.0–95.4)
Pup body weight on PND 35 (g)	826.0 ± 170.5	738.5 ± 79.7	759.3 ± 99.2	796.9 ± 127.0	750.6 (702.5–838.8)

^a^ Charles River Lyon (Test Facility) historical control data from 2013 to 2018 (fertility and delivery data) and 2018 (Caesarean section data), ^b^ Summarized data for Cohorts 1 and 2, ^c^ Calculated as number of mated females/number paired × 100, ^d^ Calculated as number pregnant/number mated × 100, ^e^ Calculated as number pregnant/number paired × 100, ^f^ One animal each in the Al(OH)_3_ control and RSVpreF + Al(OH)_3_ groups aborted and thus was excluded from these data, ^g^ Data from the one Al(OH)_3_ control litter that died on LD 5 and one RSVpreF + Al(OH)_3_ litter that died on LD 4 were both included in these delivery cohort data, resulting in slightly lower viability index and lactation index, respectively, ^h^ kit survival PND 0–4, ^i^ kit survival PND 4–35.

## Data Availability

The original contributions presented in this study are included in the article/Appendix A. Further inquiries can be directed to the corresponding author.

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
