# Peer review of "Favorable Nonclinical Safety Profile of RSVpreF Bivalent Vaccine in Rats and Rabbits"

_vaccines, 2024, doi:10.3390/vaccines13010026_

Round 1

Reviewer 1 Report

Comments and Suggestions for Authors

In their abstract and introduction the authors state that RSV is the leading cause of infant death. This is not the case: The leading causes according to WHO data  include premature birth, birth complications (birth asphyxia/trauma), neonatal infections and congenital anomalies.

In the method section the authors fail to justify the sample sizes of n=22 in groups. A sample size calculation needs to be presented.

The authors need to justify the animal models chosen.

The authors need to document the details of the Institution which granted ethical and research approval with approval number.

Author Response

In their abstract and introduction the authors state that RSV is the leading cause of infant death. This is not the case: The leading causes according to WHO data  include premature birth, birth complications (birth asphyxia/trauma), neonatal infections and congenital anomalies.

Author response: Thank you for pointing this out. We agree the statement in the abstract is an inaccurate statement as currently written. A qualifier “infectious disease” (causing infant death and hospitalization) has been added to the statement in both sections with supporting references provided in the Introduction.

In the method section the authors fail to justify the sample sizes of n=22 in groups. A sample size calculation needs to be presented.

Author response: The sample size used in the DART study is based on regulatory requirements, expectations, and experience detecting developmental and reproductive effects in rodents. At the time the DART study was conducted, it was designed to be compliant with regulatory guidelines including the 2006 FDA Guidance entitled, “Considerations for developmental toxicity studies for preventative and therapeutic vaccines for infectious disease indications” with reference to ICH S5(R2). These guidelines describe a target group size of 20 and, for the DART vaccine study design, it should allow an evaluation of at least 40 animals per group, allocated to the Caesarean and littering subgroups using 20 animals each. In addition to regulatory requirements, the justification for this group size provided in ICH S5(R2) described that for all but the rarest events, evaluation of between 16 and 20 litters for rodents and rabbits tends to provide a degree of consistency between studies. Below 16 litters per evaluation, between-study results become inconsistent; above 20-24 litters per group, the consistency and precision are not greatly enhanced. To allow for natural wastage, the starting group size should be larger. Therefore, it is common to increase the group size slightly to ensure there are sufficient number of pregnant animals per phase for the evaluations specified in these regulatory guidance documents. Based on our experience, a group size of 22 per phase per group is usually sufficient to allow a definitive interpretation of the study endpoints.

To provide insight into the above justification, we have added the following sentence and references to the methods section of the manuscript (Section 2.3.2), “This study design was designed for compliance with regulatory guidances at the time, including the number of animals per group (ICH S5R2 2005, FDA 2006)”.

The authors need to justify the animal models chosen.

Author response:   Thank you for the comment and we have added statements in the manuscript that the selection of rats and rabbits as the toxicology study species as well as the overall study design were according to relevant regulatory guidelines (with references): Sections 2.1 (2nd paragraph) and 2.3.2 (1st paragraph).

The authors need to document the details of the Institution which granted ethical and research approval with approval number.

Author response:   With the initial submission, we had supplied copies of the study protocols showing Animal Use Protocol (AUP) approval for the RDT study and Ethical Committee approval for the DART study with authorization numbers, which were deemed sufficient by the editorial office. We now have included these details in the manuscript.

Reviewer 2 Report

Comments and Suggestions for Authors

The manuscript is well written- it gives enough background, describes the method and discusses the findings with relevance. It is a straight forward study for safety with respect to RDT and DART. 

There are few points that the authors should include in the manuscript.

1. Provide limitations of this study.

2. Minor point: Did the authors perform any animal feces and urine analysis for this study? 

Overall, this manuscript can be recommended to the Editor for publication upon revision.

Author Response

Comment 1: Provide limitations of this study.

Author response: We appreciate the reviewer’s comment and have added this in the Discussion section of the revised manuscript.

Comment 2: Minor point: Did the authors perform any animal feces and urine analysis for this study? 

Author response: We did not perform analysis on animal feces but did conduct urinalysis in the RDT study. The results of the urinalysis (no effect) are mentioned in Section 3.1 (Clinical Laboratory Measurement) and data included in Supplemental Table 2.

Reviewer 3 Report

Comments and Suggestions for Authors

The manuscript describes nonclinical safety studies of the adjuvanted (Al(OH)3) anti-respiratory syncytial virus Abrysvo vaccine. No doubt that preclinical studies of the produced Abrysvo vaccine, which undergoes phase 3 clinical studies, have been successfully finished before its clinical application. Hence the necessity of the improvement of this vaccine and the safety studies of the improved variant should be highlighted in the Introduction. Otherwise the aim of the safety and efficiency studies seems unclear. The same note relates to the Discussion section and the Abstract. 

  All figure and table legends contain too much information that repeats the text of Materials and methods section and the overview of the study design presented in Fig.1. This excess information should be deleted.

 Anti-RSVproF antibody titer determination is not described and no paper is cited that describes it. 

Instead of the "data not shown" statements it is better to provide the results with nonsignificant differences as Supplementary materials.

Data presentation in Figs. 3 and 4 is somewhat difficult to understand. It would be better to use a chronological presentation as in Fig.4, it seems more clear. 

The statement about the accumulation fo macrophages (line 417) should be confirmed by the results of cell counting.      

Author Response

Comment 1: The manuscript describes nonclinical safety studies of the adjuvanted (Al(OH)3) anti-respiratory syncytial virus Abrysvo vaccine. No doubt that preclinical studies of the produced Abrysvo vaccine, which undergoes phase 3 clinical studies, have been successfully finished before its clinical application. Hence the necessity of the improvement of this vaccine and the safety studies of the improved variant should be highlighted in the Introduction. Otherwise the aim of the safety and efficiency studies seems unclear. The same note relates to the Discussion section and the Abstract. 

Author response: We certainly agree with the Reviewer’s comments on the importance of nonclinical studies in vaccine development, but we are not sure that we fully understand the latter part of the comment because the aims of the study are stated in the Introduction and concluded in the Discussion section. The RSVpreF Bivalent Vaccine as reported in this article is the same vaccine as what is in Abrysvo®; and the RDT and DART studies were conducted to “to support the clinical development and eventual licensure of the vaccine candidate for the adult and maternal indications”. It is not a newer vaccine than Abrysvo®.

Comment 2: All figure and table legends contain too much information that repeats the text of Materials and methods section and the overview of the study design presented in Fig.1. This excess information should be deleted.

Author response: Thank you for pointing this out. We have removed or significantly shortened redundant information from the figure or table legends.

Comment 3: Anti-RSVproF antibody titer determination is not described and no paper is cited that describes it. 

Author response: Details of immunogenicity evaluation is described in Section 2.4 and antibody measurements were described in the last two paragraphs with a reference #28 (in the supplemental material of the referenced article) cited for more details in the technical aspects. We now added “(Titer was calculated) as the interpolated reciprocal of the serum dilution resulting in 50% reduction in the number of viral focus forming units when compared to the control without test serum” to the manuscript.

Comment 4: Instead of the "data not shown" statements it is better to provide the results with nonsignificant differences as Supplementary materials.

Author response: We now have supplied all the data that were previously marked as “data not shown” as supplemental materials except for ophthalmological results (Section 3.1, end of “In-Life Observations”), because its data table would be simply “no abnormal findings” for each group.

Comment 5: Data presentation in Figs. 3 and 4 is somewhat difficult to understand. It would be better to use a chronological presentation as in Fig.4, it seems more clear. 

Author response: We appreciate the feedback and agree that Fig. 4 has data presented clearly, with males and females separated for each group and each timepoint laid out chronologically. However, Fig. 3 only shows data at two timepoints for the maximal effect of the test articles, one at the end of dosing phase (D38) and the other at the end of the recovery phase (recD28/recD27), which are indeed presented chronologically. By using different colors, the two sexes can be differentiated easily.   

Comment 6: The statement about the accumulation fo macrophages (line 417) should be confirmed by the results of cell counting.      

Author response:

We greatly appreciate the feedback from the reviewer. While the technique (microscopic evaluation) used in these experiments for illustrating the accumulation of inflammatory cells at the vaccine administration site was qualitative, confidence in the identification of macrophages is high, as the assessment was performed by an experienced board-certified veterinary pathologist. Macrophages are readily identifiable microscopically and are shown in the manuscript images. Assessment of the inflammatory cell infiltrates at the injection sites and draining lymph nodes by cell counting via technologies as flow cytometry could certainly give some insight into the actual infiltrating cell types, cell subtypes, and cell numbers. However, for the purposes of regulated GLP repeat-dose toxicity studies, such measurements would add little value to the overall conclusions related to safety. Microscopic assessment of injection sites in vaccine regulatory studies for evaluating local inflammatory effects of vaccine administration is the standard of practice, and consistent with relevant WHO guidelines as cited in the manuscript. Additionally, the use of only microscopic assessment of the vaccine administration sites and draining lymph nodes is consistent with similar publications on vaccine safety. As examples, please see our prior publication as well as a publication by others.

1) “Evaluation of Rat Acute Phase Proteins as Inflammatory Biomarkers for Vaccine Nonclinical Safety Studies” (https://journals.sagepub.com/doi/10.1177/0192623320957281)

2) “Nonclinical safety assessment of repeated administration and biodistribution of a novel rabies self-amplifying mRNA vaccine in rats” (https://pubmed.ncbi.nlm.nih.gov/32240713/ )

Reviewer 4 Report

Comments and Suggestions for Authors

The authors have presented a well-planned and well-designed experimental study. The methods used and the quality of the results obtained give no cause for complaint. Based on the results of the study, the authors concluded: " the nonclinical safety evaluation of RSVpreF, with or without Al(OH)3, demonstrated a tolerable nonclinical safety profile supporting clinical development and licensure/marketing authorization", which is directly related to the purpose of the study. I do not see the need for significant modernization of this high-quality scientific work. In the meantime, I suggest that the authors make minor changes to the text of the article:

(1) Line 125. "Sodium chloride" should be added to 0.9%.

(2) References need to be adapted to MDPI style.

Author Response

The authors have presented a well-planned and well-designed experimental study. The methods used and the quality of the results obtained give no cause for complaint. Based on the results of the study, the authors concluded: " the nonclinical safety evaluation of RSVpreF, with or without Al(OH)3, demonstrated a tolerable nonclinical safety profile supporting clinical development and licensure/marketing authorization", which is directly related to the purpose of the study. I do not see the need for significant modernization of this high-quality scientific work. In the meantime, I suggest that the authors make minor changes to the text of the article:

Author response: We greatly appreciate the reviewer’s remarks.

(1) Line 125. "Sodium chloride" should be added to 0.9%.

Author response: It has been added.

(2) References need to be adapted to MDPI style.

Author response: We have converted the reference style to MDPI style.

Round 2

Reviewer 1 Report

Comments and Suggestions for Authors

The authors have addressed my comments adequately.

Author Response

Thank you again for your critical review!

Reviewer 3 Report

Comments and Suggestions for Authors

Regarding the author's comment 1. Since ABRYSVO is a vaccine registered for clinical use, its nonclinical testings have been already performed. Does the manuscript present the results of these previous testings (which were performed before the vaccine was registered and were not published earlier)? Or it discloses the results of testing an improved, i.e. adjuvanted version of the vaccine? It seems unclear why toxicology and efficacy testing results of the vaccine are presented for the publication AFTER its registration for clinical use. However, it is well understood if the adjuvanted vaccine is compared to the nonadjuvanted one, in order to demonstrate the same safety and a higher efficacy. To my mind, it should be made clear in the Introduction and Discussion sections.

Comments 2-4. Agreed.

Comment 5. I don't insist on my viewpoint. However, too small graphs and the proximity of datapoints can really complicate the understanding of these figures. Please check the graph for A/G ratio: is the double asterisk put correctly? It seems that it should mark the other datapoint, or more datapoints.

Comment 6. Agreed.   

Author Response

Reviewer's Comments: Regarding the author's comment 1. Since ABRYSVO is a vaccine registered for clinical use, its nonclinical testings have been already performed. Does the manuscript present the results of these previous testings (which were performed before the vaccine was registered and were not published earlier)? Or it discloses the results of testing an improved, i.e. adjuvanted version of the vaccine? It seems unclear why toxicology and efficacy testing results of the vaccine are presented for the publication AFTER its registration for clinical use. However, it is well understood if the adjuvanted vaccine is compared to the nonadjuvanted one, in order to demonstrate the same safety and a higher efficacy. To my mind, it should be made clear in the Introduction and Discussion sections.

Author response: While it is more ideal that nonclinical testing results of a commercial vaccine are published prior to the market authorization, this is not always possible. Practical reasons within a company such as concerns about disclosing proprietary information, waiting for confirmatory clinical data, legal clearance, and complicated approval processes, etc. can often delay their publishing. It is therefore not uncommon that nonclinical data are published later from companies that have already obtained market approval for a product. Due to high interest in vaccine safety, not only from the scientific community but in general public, we feel a full disclosure of RSVpreF nonclinical safety data remains important. This manuscript provides additional details, as compared with the key nonclinical safety testing information available at the regulatory websites.

We also feel that we had made it clear in the last paragraph of the Introduction that Abrysvo is RSVpreF (1st sentence), which was followed by how it was discovered (construct 847 selected) and the nonclinical studies conducted on RSVpreF (847A + 847B). In the Discussion, we linked the nonclinical safety of RSVpreF to Abrysvo clinical data and concluded that the profile supported its development and market authorization.

It might also be worth mentioning that the adjuvanted version of RSVpreF, RSVpreF + Al(OH)3), was an additional candidate that was evaluated, side by side with RSVpreF, in the early nonclinical and clinical development. However, it was determined in clinical testing that RSVpreF alone produced a sufficient, antigen-specific, neutralizing antibody response. Thus, RSVpreF alone was the final commercial formulation of the vaccine for both maternal immunization and adults. This point is made in the first paragraph of the Discussion section.

Comment 5. I don't insist on my viewpoint. However, too small graphs and the proximity of datapoints can really complicate the understanding of these figures. Please check the graph for A/G ratio: is the double asterisk put correctly? It seems that it should mark the other datapoint, or more datapoints.

Author’s response: We greatly appreciate the Reviewer’s comment and have made the figures larger by using one set of figure keys instead of repeating them. We also added dotted lines between groups so that data between groups can be easily discerned. As explained in the figure legend, data in RSVpreF or RSVpreF + Al(OH)3 groups are compared with those in saline or Al(OH)3 control at each time point, respectively. The only statistical significances are at the end of dosing phase (Day 38) for males and/or females in the RSVpreF Al(OH)3 group, and we confirm that the A/G ratio data are correct - both male and female Day 38 values are significantly different with p<0.01 as compared with the Al(OH)3 control (data were QC’d).